# Deep Reinforcement Learning from Hierarchical Preference Design

Alexander Bukharin [1]    Yixiao Li [2]    Pengcheng He [3]    Tuo Zhao [2]

## Abstract

Reward design is a fundamental, yet challenging aspect of reinforcement learning (RL). Researchers typically utilize feedback signals from the environment to handcraft a reward function, but this process is not always effective due to the varying scale and intricate dependencies of the feedback signals. This paper shows by exploiting certain structures, one can ease the reward design process. Specifically, we propose a hierarchical reward design framework – HERON for scenarios: (I) The feedback signals naturally present hierarchy; (II) The reward is sparse, but with less important surrogate feedback to help policy learning. Both scenarios allow us to design a hierarchical decision tree induced by the importance ranking of the feedback signals to compare RL trajectories. With such preference data, we can then train a reward model for policy learning. We apply HERON to several RL applications, and we find that our framework can not only train high performing agents on a variety of difficult tasks, but also provide additional benefits such as improved sample efficiency and robustness.

## 1. Introduction

Over the past decade, significant advancements in deep learning techniques, along with unprecedented growth in computational power, have facilitated remarkable achievements in the field of deep reinforcement learning (RL) across diverse domains, including finance, transportation, and automatic programming (Deng et al., 2016; Haydari & Yılmaz, 2020; Le et al., 2022). A key component of modern RL is the reward function, which is typically predefined in benchmark environments such as the OpenAI gym or games (Mnih et al., 2013; Silver et al., 2016; Brockman et al., 2016). When dealing with complex real-world environments, however,

we are unable to access to the ground-truth reward, or the reward is sparse: we receive zero reward most of the time. Therefore, designing a reward for the agent is necessary.

To construct the reward, practitioners often use multiple feedback signals $z_1, \ldots, z_m$, each of which captures different facets of an agent's behavior. In settings where the reward is inaccessible, the most common approach to utilizing these signals is the linear combination, e.g., $r = \sum_i \omega_i * z_i$ (Booth et al., 2023; Le et al., 2022; Zhang et al., 2019). The hyperparameters $\omega_i$'s are tuned to provide a comprehensive description of the agent's behavior, a process commonly known as reward engineering (Fu et al., 2017; Wu et al., 2021). Taking the traffic light control as an example, Zhang et al. (2019) consider a weighted combination of vehicle queue length, average waiting time and some other feedback signals as the reward. The hyperparameters $\omega_i$'s are determined by extensive tuning in Van der Pol & Oliehoek (2016); Wu et al. (2017); Zhang et al. (2019). In sparse reward settings, the feedback signals are used to compose a reward surrogate, which is then combined with the sparse reward. Taking code generation as an example, Le et al. (2022) design a piece-wise function where the region is divided according to the feedback signals (e.g., compilation error, runtime error) and values are designed by human experts.

Although feedback signals may serve as useful criteria of an agent's behavior, reward engineering is not always an effective way to ensemble these signals. This is because feedback signals may have different scales as well as intricate dependencies with other feedback signals. In this case, determining the weight by humans becomes challenging, and multiple weights must be simultaneously tuned since their respective feedback signals are often correlated. Moreover, in sparse reward settings, a piece-wise reward function often requires massive trials to determine the value, and dividing the region is also difficult because of the complex relation among different feedback signals.

How to address the aforementioned issues still remains unclear. While most existing works have focused on reward engineering for specific applications (Liu et al., 2020; Zhang et al., 2019), this paper proposes a novel reward design framework for when feedback signals exhibit hierarchical relations. This relation exists in many RL problems. For

---

*Equal contribution [1]NVIDIA [2]Georgia Institute of Technology [3]Zoom Communications. Correspondence to: Alexander Bukharin <abukharin@nvidia.com>.

*Proceedings of the 42nd International Conference on Machine Learning*, Vancouver, Canada. PMLR 267, 2025. Copyright 2025 by the author(s).

example, in traffic light control the vehicle queue length significantly outweighs the average wait time and other feedback signals. Our framework is also suitable for sparse reward settings, where sparse rewards naturally have greater importance than the surrogates. For code generation, practitioners enhance the sparse reward (whether the code passes unit tests) with surrogates, such as the type of error.

To leverage such hierarchical structures in the aforementioned scenarios, we propose HERON (**H**ierarchical prEference-based **R**einf**O**rcement lear**N**ing). HERON trains a preference-based reward model (Bradley & Terry, 1952; Ouyang et al., 2022) through pair-wise trajectory comparisons. Specifically, we design a decision tree, at each level of which compares trajectories based on a feedback signal. The feedback signal we use at each level is determined by its importance ranking, as assigned by the human annotator.

This decision tree based on importance ranking provides a number of benefits. First, it is a more natural way to resemble the human decision process compared to reward engineering. When making decision between two choices, humans typically start with the the most important factor, then proceed to the secondary factor, and continue until the remaining less important factors. Second, ranking feedback signals is usually easier than specifying numerical weights. Third, the comparison process of HERON does not depend on the absolute value of a feedback signal, but on their relative quantity. We find that this brings additional robustness in scenarios where the training environment changes. We will further discuss this in Section 4.1. Finally, HERON is able to leverage pre-trained knowledge, which allows for the creation of more powerful rewards.

We empirically validate HERON framework through extensive experiments on real world applications:

**Traffic Light Control.** In traffic light control (Zhang et al., 2019), there are 6 feedback signals with the hierarchy: queue length > the average vehicle waiting time > other feedback signals. HERON consistently outperforms the policies trained with reward engineering techniques.

**Code Generation.** Code generation (Le et al., 2022) is a sparse reward scenario. Most of the programs generated by the policy cannot pass all the unit tests and thus fail to receive the reward. Therefore, surrogates, like the type of error, are added to compose a piece-wise reward function. In code generation, HERON demonstrates the ability to achieve higher Pass@$K$ scores compared to the hand-crafted piece-wise reward function employed in state-of-the-art approaches.

**Language Model Alignment.** Although language models are powerful, they are not always aligned with human principles (Brown et al., 2020). We propose to use HERON to align language models, by using public datasets of language

model prompts and outputs labelled with response helpfulness, coherence, correctness, verbosity, and complexity. By ranking these factors and applying HERON, we are able to train an aligned language model.

**Robotic Control.** We also evaluate HERON on robotic control (Coumans & Bai, 2016; Brockman et al., 2016), where the hierarchy of the feedback signals is unclear. In these environments, HERON performs better than reward engineering and even achieves comparable performance compared to the ground-truth reward. This shows that HERON is able to train a reasonable policy even if the hierarchy is unclear.

The rest of this paper is organized as follows: Section 2 introduces the related work. Section 3 introduces our proposed reward design framework. We conduct experiments in Section 4. We discuss the limitations of our method in Section 5.

## 2. Related Work

Besides reward engineering, there are several works that attempt to improve reward design for RL.

**Reward Shaping.** Reward shaping aims to accelerate the convergence of RL algorithms by incorporating auxiliary reward information through shaping functions (Ng et al., 1999; Tenorio-Gonzalez et al., 2010; Devlin & Kudenko, 2012). These approaches aim to mitigate the sparsity of a pre-defined reward function. While reward shaping has demonstrated success in practice, it often necessitates extensive tuning. To circumvent the need for costly tuning, several methods have been proposed to automatically shape rewards by utilizing an abstract MDP (Marthi, 2007), tile coding (Grzes & Kudenko, 2008), and bi-level optimization (Fu et al., 2019; Hu et al., 2020). In contrast, our work pursues a different direction that eliminates the requirement for a pre-specified reward function and does not assume that the reward is a linear combination of auxiliary factors.

**AutoRL.** AutoRL (Afshar et al., 2022; Parker-Holder et al., 2022) automates various aspects of hyperparameter selection in RL, including parameters related to the reward. Particularly relevant to our work, Faust et al. (2019) and Chiang et al. (2019) treat reward weights as hyperparameters and optimize them using population-based training.

**Inverse Reinforcement Learning.** Inverse reinforcement learning (IRL) aims to learn a reward function from expert demonstrations (Ng et al., 2000; Abbeel & Ng, 2004; Boularias et al., 2011). Although IRL enables the learning of complex behaviors without manual reward tuning, it requires observed, optimal behavior. These demonstrations are often costly to obtain, and in our experiments, acquiring them would be far more expensive than obtaining a hierarchy of feedback signals. Furthermore, IRL methods

typically require unstable bi-level optimization procedures, which our approach does not involve.

**Reinforcement Learning from Human Feedback.** Reinforcement learning from human feedback (RLHF) (Christiano et al., 2017; Ouyang et al., 2022; Bai et al., 2022) aims to train a policy model that aligns with human preference. Although both RLHF and our method involve a preference-based reward model, in RLHF the preference labels come directly from the annotators. On the other hand, HERON automatically compares trajectories by the importance ranking of the feedback signals. This ranking can be easily set up by a human overseer if the feedback signals have hierarchy, or in sparse reward settings. We will discuss more differences in Section 5.

## 3. Method

We consider a Markov decision process $\langle \mathcal{S}, \mathcal{A}, \mathcal{P}, \mathcal{R}, \gamma \rangle$ where an agent interacts with an environment over a series of discrete time steps. At time step $t$, the agent observes $s_t \in \mathcal{S}$, takes an action $a_t \in \mathcal{A}$ according to a policy, and receives the next state observation $s_{t+1} \in \mathcal{S}$ and reward $r_t \in \mathbb{R}$. In most real-world applications, the reward $r_t$ is not available. Therefore, our goal is to design an appropriate reward model $R_\phi : \mathcal{S} \times \mathcal{A} \to \mathbb{R}$ such that when maximizing the objective

$$J(\theta) = \mathbb{E}_{\tau \sim \pi_\theta} \left[ \sum_{(s_t, a_t) \in \tau} \gamma^t R_\phi(s_t, a_t) \right], \qquad (1)$$

the agent's behavior, guided by the policy model $\pi_\theta : \mathcal{S} \times \mathcal{A} \to \mathbb{R}$, meets our expectation.

To design the reward, we utilize a set of $n$ feedback signals $z_t^1, \ldots, z_t^n$ given at each time step $t$. These signals serve as multiple measurements of a trajectory. We denote segments of the resulting trajectory as

$$\tau = (s_t, a_t, \{z_t^i\}_{i=1}^n), \ldots, (s_{t+k}, a_{t+k}, \{z_{t+k}^i\}_{i=1}^n),$$

and we overload the notation for $z_i$ such that it represents the feedback signal of a segment of trajectory, $z_i(\tau) = \sum_{(s_t, a_t) \in \tau} z_i(s_t, a_t)$.

Given the feedback signals of trajectories and the importance ranking, HERON builds the preference-based reward model by *preference elicitation* and *reward learning*. HERON then trains a policy model through *policy learning*.

**Preference Elicitation.** We generate a set of trajectory data with a policy model. We can obtain the initial policy model by (i) behavior cloning from expert demonstration data, (ii) pre-training it using a handcrafted reward, or (iii) purely random initialization. With the trajectory data, HERON first compares them based on an intuitive form of domain

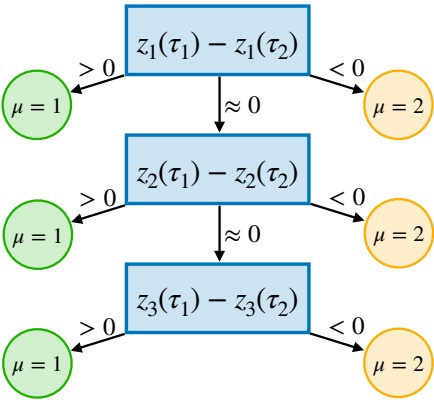

*Figure 1.* Preference elicitation: compare trajectories $\tau_1$ and $\tau_2$ through 3 feedback signals $z_1, z_2, z_3$.

knowledge: rankings over the feedback signals. We assume $z_1, \ldots, z_n$ have been ordered in descending order of importance by an expert with domain knowledge. In sparse reward settings, $z_1$ is always the sparse reward and the remaining $z_i$'s are the surrogates. We then elicit a preference $\mu \in \{0, 1, 2\}$ between trajectory pairs $(\tau_1, \tau_2)$ with a decision tree induced by the given feedback signal hierarchy. A tie is denoted by $\mu = 0$, $\mu = 1$ means $\tau_1$ is preferred, and $\mu = 2$ means $\tau_2$ is preferred.

The decision tree is constructed as follows. We first set the current level $l = 1$. We then calculate

$$\mu = \begin{cases} 0 & \text{if } |z_l(\tau_1) - z_l(\tau_2)| \le \delta_l \\ 1 & \text{if } z_l(\tau_1) > z_l(\tau_2) + \delta_l \\ 2 & \text{if } z_l(\tau_2) > z_l(\tau_1) + \delta_l \end{cases}$$

where $\delta_l$ is a margin hyperparameter for level $l$. The margin parameter $\delta_l$ is ensures that we only elicit a preference using $z_l$ if the two trajectories are significantly different according to $z_l$. The margin $\delta_l$ can be used to inject further domain knowledge into the HERON algorithm, but in our experiments we set $\delta_l$ to the standard deviation of $z_l$ over the collected data.

If $\mu = 0$, we update $l \leftarrow l + 1$ and compare the trajectories with the next most important feedback signal. If the two trajectories are not significantly different in any of the feedback signals (i.e. $l > n$), we discard the trajectory pair. We illustrate the algorithm in Figure 1.

**Reward Learning.** Given a labeled dataset $D$ of trajectories $(\tau_w, \tau_u)$ where $\tau_w$ is the trajectory preferred by the preference elicitation algorithm (i.e. $\mu = 1$), we would like to assign a higher reward to the preferred trajectory (we remove all ties from the dataset, since we find including them has negligible effect on training). To accomplish this, we train a reward model $R_\phi : \mathcal{S} \times \mathcal{A} \to \mathbb{R}$ where $R_\phi(\tau) = \sum_{(s_t, a_t) \in \tau} \gamma^t R_\phi(s_t, a_t)$. To assign a higher reward to the preferred trajectory $\tau_w$, we follow the methodol-

ogy in Ouyang et al. (2022) and optimize the loss

$$\mathcal{L}(\phi) = -\mathbb{E}_{(\tau_w, \tau_u) \sim D} \left[ \log \left( \sigma (R_\phi(\tau_w) - R_\phi(\tau_u)) \right) \right]. \quad (2)$$

We remark that this loss employs the Bradley-Terry preference model (Bradley & Terry, 1952). Once we have trained the reward model $R_\phi$, we can assign a reward to each trajectory $\tau$ as $R_\phi(\tau)$. It is important to note that unlike some prior works which learn linear reward models on top of state features, HERON allows for more complicated reward models parameterized by neural networks (Sadigh et al., 2017; Bıyık et al., 2020). Therefore, it is also possible to introduce pre-trained knowledge into the reward model.

**Policy Learning.** With the reward model $R_\phi$, the policy in (1) can be learned via popular reinforcement learning algorithms such as Q-learning or Proximal Policy Optimization (Sutton & Barto, 2018; Schulman et al., 2015; 2017).

**Optional: Multi-stage Training.** The success of the reward learning depends on the quality of the trajectories generated by the policy model, but if the initial policy model is not optimal, e.g., pre-trained from handcrafted reward function or randomly initialized, it may introduce significant sampling bias to the trajectories. To address this issue, we can repeat the preference elicitation, reward learning, and policy learning for multiple rounds. In each new round, trajectories in preference elicitation are generated by the policy model from the last round, and the reward model is then adapted using the new comparisons.

**Extension: Direct Preference Optimization.** Rafailov et al. (2024) showed that in contextual bandit settings, the KL-regularized preference optimization problem can be optimized directly, without a reward model. In appropriate settings, we can use DPO to simultaneously train the policy and reward, reducing the computational cost of HERON.

## 4. Experiment

We evaluate the efficacy of our framework traffic light control experiments, code generation, language model alignment, and robotic control.

### 4.1. Multi-Agent Traffic Light Control

**Environments.** In this real-world scenario, cooperative agents learn to increase the throughput and minimize the wait of cars passing through a traffic network. Due to the complexity of the traffic system, unfortunately, there is no one optimal reward, and different reward may be preferred in different scenarios. To solve this problem, Wei et al. (2018); Van der Pol & Oliehoek (2016); Zhang et al. (2019) define the ground-truth reward by balancing the following six feedback signals: queue length ($q$), vehicle waiting time ($wt$), vehicle delay ($dl$), number of vehicle emergency stops ($em$), number of light phase changes ($fl$), and number of

vehicles passing through the system ($vl$).

For these experiments, we evaluate a variety of reward hierarchies. Our reward model is parameterized by a three-layer MLP that is learned by multi-stage training. We use QCOMBO (Zhang et al., 2019), a Q-learning based algorithm as the RL algorithm and conduct experiments using the Flow framework (Wu et al., 2017). We train Multi-agent RL policies on a two-by-two grid (four agents), each parameterized by a three-layer MLP. For more details on the environment and the experiment setting, see Appendix E.

**Baseline.** We compare our method to reward engineering, where the reward is formulated as $\sum_{i=1}^{n} W_i z_i$, where the $W_i$'s are hyperparameters and $z_i$'s are normalized feedback signals. To inject HERON's domain knowledge and to make hyperparameter tuning tractable, we set the $W_i$'s to be geometrically decreasing such that $W_i = \beta^i$ and select $\beta \in \{0.1, 0.2, ..., 1.0\}$. This a very realistic and competitive reward engineering baseline. Note that we also explore reward engineering without this prior hierarchical knowledge. We also compare to ensemble approaches, which train a separate policy on each feedback signal and then select an action at each time step by a weighted combination of each policy (Brys et al., 2017).

In this experiment we evaluate different reward designs by comparing the ground-truth reward of the associated policy.

**Results.** In Figure 2, we plot the evaluation reward (as defined in (Zhang et al., 2019)) of policies in the traffic light control environment. We observe that the policy trained with HERON performs significantly better than the policies trained with the reward engineering baseline or even by the ground-truth reward developed in Zhang et al. (2019). The gain of HERON over all other methods passes a t-test with $p < 0.005$. We hypothesize that HERON can utilize each reward signal better than a linear combination does; a significant change in a single feedback signal may be drowned out in the linear combination, but HERON can incorporate this information due to its hierarchical nature.

**Flexibility of Hierarchical Reward Modeling.** In various tasks, there is no one ideal reward, and the aspects of an agent's behavior that should be prioritized depend on the practitioner's preference. As a result, a crucial characteristic that reward design algorithms should possess is flexibility. In particular, modifying the domain knowledge inputted should result in corresponding changes in the behavior of the agent. To evaluate the flexibility of HERON, we examine how changing the feedback signal rankings changes agent behavior in the traffic light control environment.

In this experiment we always set the most important signal as the number of cars passed, and then we use the queue length, wait time, or delay as the second signal. The results can be seen in Figure 4. We observe that HERON is quite

flexible, and that by changing the reward hierarchy we can significantly influence the agent's behavior: when prioritizing certain signals the policies performance (measured according to the prioritized signal) will greatly increase.

**Signal Utilization.** We also show the level the decision tree induced by HERON reaches in Figure 3. This may change with different reward hierarchies, but as we can see from the figure, a relatively similar proportion of decisions are made at each level of the decision tree. This confirms the efficacy of setting $\delta_l$ to $z_l$'s standard deviation.

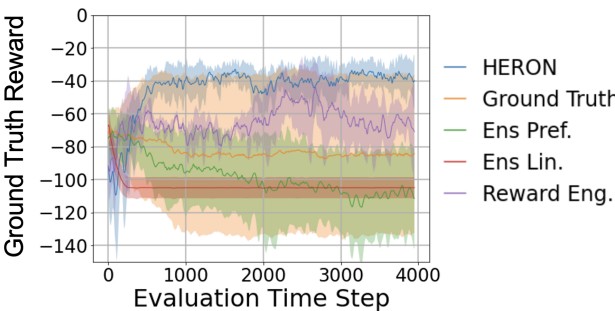

Figure 2. Evaluation curves with different reward hierarchies in traffic light control. The curve is within $\pm$ one standard deviation.

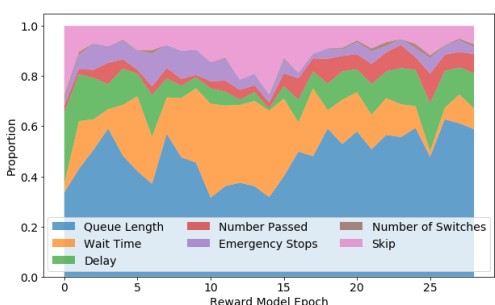

Figure 3. Utilization of different signals.

**Robustness.** An advantage of HERON is that unlike reward engineering, it does not depend on the magnitude of the different feedback signals. This is because the preference elicitation algorithm will label trajectory pairs with $\mu \in \{0, 1, 2\}$, regardless of the scale of the different signals. This scale-invariance is beneficial, since algorithms that depend on the scale of the feedback signals may be vulnerable to changes in the environment during training. For example, if the scale of a feedback signal suddenly doubles, (i.e. the traffic on a highway doubles due to rush hour) then two things will happen: (1) the scale of the reward signal may sharply increase, which is similar to a sudden change in learning rate; (2) the weight vector used in reward engineering to combine the feedback signals will effectively be changed. The first phenomenon may cause training instability, and the second phenomenon could cause the agent to

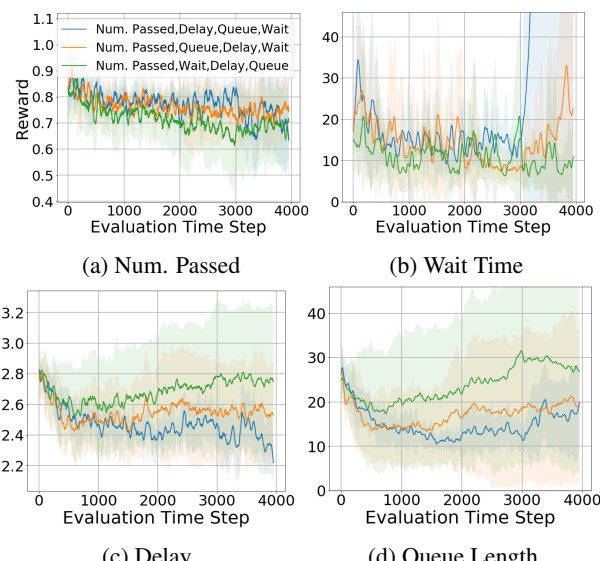

Figure 4. Evaluation reward curves with different reward hierarchies in traffic light control. The importance decreases from left to the right in a label. The curve is within $\pm$ one standard deviation.

be misaligned with the human overseer's desires.

To evaluate HERON's robustness, we change the speed of the cars halfway into training (this a realistic setting, since many areas have time-dependent speed limits). We then evaluate each policy after training under the new environment, and see which algorithms were able to adapt the best. We compare the HERON-trained policy with two policies trained with the reward engineering: one that uses the optimal learning rate in the unchanged environment ($1 \times 10^{-3}$) and one that uses a smaller, more stable learning rate of $1 \times 10^{-5}$.

From Figure 5, we can see that reward engineering is quite sensitive to changes in the environment during training. This can be combatted with a smaller learning rate, but this will result in slower learning and a sub-optimal reward. On the other hand, HERON is able to attain a high reward regardless of the environment change, supporting our hypothesis that HERON's scale-invariant design leads to increased robustness.

### 4.2. Code Generation

**Environment.** RL has recently gained considerable attention for its state-of-the-art performance in various text generation tasks. Therefore, we investigate if HERON can achieve similar improvements in LLM performance solely based on rankings over feedback signals. First, we consider the code generation task. In code task, the goal of the agent is to write a program that will satisfy the criteria specified in a given problem.

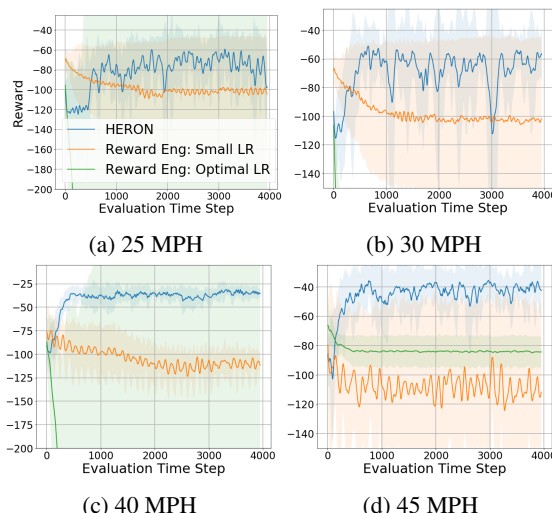

*Figure 5.* Evaluation curves with different environments: changes of vehicles' speed limit. The baseline speed limit is 35 MPH.

**Baselines.** Recently, Le et al. (2022) demonstrated state-of-the-art performance can be achieved by training with RL. They manually design a constant piece-wise reward that is determined by feedback signals including whether the program passes all the unit test and the type of error if failed (i.e. compilation error or runtime error).

$$R_{\text{CodeRL}}(s) = \begin{cases} -1.0 & \text{if program } s \text{ fails to compile} \\ -0.6 & \text{if program } s \text{ has a runtime error} \\ -0.3 & \text{if program } s \text{ fails a unit test} \\ +1.0 & \text{if program } s \text{ passes all unit tests.} \end{cases}$$

Shojaee et al. (2023) (PPOCoder) build upon this work, integrating more feedback signals such as a program's abstract syntax tree (AST) similarity to expert demonstrations.

**Implementation Details.** Our decision tree is based on three signals: the percent of tests a program passes, the type of error a program incurs, and the AST similarity to expert demonstrations. To train policies we follow the implementation of Le et al. (2022). We initialize our policies with the CodeT5plus-large model and our reward model with CodeT5-small (Wang et al., 2021). The policies are first trained with behavior cloning on the expert demonstrations. Next, we generate 20 samples per training program, and conduct RL training over these generated samples. We train with the policy gradient objective. We evaluate the performance of each algorithm using Pass@K metric, which is the number of programming problems passed with K submissions per problem (Chen et al., 2021). We primarily evaluate HERON on APPS, a python programming datasets containing 5000 test problems (Hendrycks et al., 2021). Each question in the dataset comes with expert demonstrations and test cases the program should pass. To evaluate each algorithm, we generate 200 programs per problem. In total,

each method is evaluated on 1 million generated programs. To evaluate the generalization ability of the policies, we evaluate each policy in a zero-shot manner on the MBPP dataset, which contains 974 basic python programming questions (Austin et al., 2021).

**Post-Training Reward Scaling**. To further incorporate domain knowledge and environment feedback into the reward, we propose to rescale the reward learnt from (2). Specifically, we multiply the reward $R_\phi(\tau)$ by a shaping constant, denoted as $\alpha^{F(\tau)}$. Here, $\alpha$ is a hyperparameter and tuned over $\{1, 2, 3\}$, while $F(\tau)$ corresponds to a piece-wise function of the feedback signals. We define it for code generation as

$$F(\tau) = \begin{cases} 3 & \text{if program } \tau \text{ passes all unit tests} \\ 2 & \text{if program } \tau \text{ fails a unit test} \\ 1 & \text{if program } \tau \text{ yields any error.} \end{cases}$$

This function is motivated by the importance ranking of feedback signals. It explicitly reinforces feedback signals in policy learning according to their importance ranking and serves as the supplement of the preference-based reward model. Specifically, by tuning $\alpha$, we can effectively control the reward's shape and the degree of separation between the best and worst trajectories. We focus our $\alpha$ tuning efforts exclusively on the code generation task due to its high complexity.

**Results.** We display the results for the code generation task in Table 1 and Table 2. HERON outperforms all other approaches. For larger $K$ in Pass@K the gain in Pass@K pass a t-test with $p < 0.05$. This is most likely because reward engineering only gives a large reward to programs that pass all the unit tests or are similar to the expert demonstrations, while HERON can give a large reward to programs that may fail some unit tests but the reward model predicts as being likely to satisfy the prompt. This means that HERON will promote a more diverse set of programming strategies. In addition, HERON's smooth reward function (opposed to the discontinuous piece-wise function in sparse reward settings) may be more conducive for learning, and therefore lead to higher performance.

We further analyze code generation performance using the filtered Pass@$K$ metric, which only submits programs that pass unit tests provided in the prompt (Chen et al., 2021). As seen in Table 3, HERON uniformly and significantly outperforms the baselines, confirming the efficacy of HERON.

As in Le et al. (2022), we evaluate the performance of policies trained by HERON on the MBPP dataset. The results are displayed in Table 4. HERON outperforms the other methods, confirming its efficacy.

*Table 1.* Raw Pass@K on APPS.

|  | Pass@1 | Pass@5 | Pass@10 | Pass@20 | Pass@50 |
|---|---|---|---|---|---|
| BC | 1.59 | 3.82 | 5.19 | 6.74 | 6.74 |
| CodeRL | 1.71 | 4.12 | 5.57 | 7.26 | 9.81 |
| PPOCoder | 1.23 | 3.08 | 4.19 | 5.50 | 7.62 |
| HERON | **1.72** | **4.19** | **5.71** | **7.49** | **10.19** |

*Table 2.* Pass@50 on APPS by three difficulty levels: introductory, interview, competition.

|  | Intro. | Inter. | Comp. |
|---|---|---|---|
| BC | 18.8 | 4.23 | 2.10 |
| CodeRL | 23.7 | 6.93 | 4.51 |
| PPOCoder | 18.6 | 5.41 | 3.23 |
| HERON | **24.6** | **7.28** | **4.53** |

*Table 3.* Filtered Pass@K on APPS.

|  | Pass@1 | Pass@10 | Pass@20 |
|---|---|---|---|
| BC | 4.70 | 6.36 | 6.44 |
| CodeRL | 5.73 | 8.57 | 8.96 |
| PPOCoder | 5.60 | 8.61 | 8.93 |
| HERON | **5.74** | **9.03** | **9.43** |

*Table 4.* Pass@K on MBPP

|  | Pass@1 | Pass@2 | Pass@5 |
|---|---|---|---|
| CodeRL | 6.58 | 10.27 | 16.24 |
| PPOCoder | 6.58 | 10.09 | 15.85 |
| HERON | **7.40** | **11.03** | **16.54** |

## 4.3. Language Model Alignment

**Environment.** Beyond code generation, we also evaluate the ability of HERON to train instruction following models that are aligned with human principles. For this experiment we employ Phi-2 (Javaheripi et al.) as our base model, and train it on the HelpSteer dataset (Wang et al., 2023). The HelpSteer dataset is composed of instruction-response pairs annotated (from 0-4) across five feedback signals: correctness, helpfulness, coherence, complexity, and verbosity.

**Implementation.** For HERON, we use the following hierarchy: correctness > helpfulness > coherence > complexity > verbosity. We then use HERON-DPO to optimize the policy. We consider two reward engineering baselines: REINFORCE-based finetuning with equal reward weights on all signals and REINFORCE-based finetuning with exponential decaying reward weights (Williams, 1992). Every algorithm is initialized from a version of Phi-2 that has undergone supervised finetuning on HelpSteer.

**Results.** To evaluate the resulting models, we use three state-of-the-art reward models (RLHF-Flow (Dong et al., 2024), PairRM (Jiang et al., 2023), Eurus-7B (Yuan et al., 2024)) and Claude 3 Sonnet to compute a win-rate compared to the SFT policy on the HelpSteer test dataset. The results can be seen in Table 5. We find that HERON-DPO can significantly outperform the baselines across all judges. We hypothesize that this gain is due to the fact that HERON's decision tree over these signals can better capture human principles compared to linear combinations of feedback signals. More details can be found in Appendix M.

## 4.4. More Experiments

To demonstrate that HERON is able to train a reasonable policy even if the hierarchy is unclear, we experiment on four robotic control tasks: Ant, Half-Cheetah, Hopper, and Pendulum. We use the PyBullet simulator, where the ground-truth reward is formulated as a linear combination of several signals such as the robot's potential, the power cost, whether the joints are at their limit, and whether the robot's feet collide (Coumans & Bai, 2016). These factors do not necessarily display a clear hierarchy. More details on this environment can be found in Appendix K. The results can be found in Table 6, where we observe that HERON can always exceed the performance of the reward engineering baseline.

## 4.5. Ablation

**Training Time Analysis.** The main computational cost of HERON comes from reward model training, as data collection is already a part of most RL algorithms and preference elicitation is very fast. To accelerate reward model training in the multi-stage setting, we can use an annealed training schedule (see Appendix I). The normalized training time of HERON, reward engineering, and ensemble-based learning are shown in Figure 6a. HERON is 25% slower than reward engineering on average, which is quite reasonable given that the tuning cost of reward engineering is usually large.

**Hyperparameters.** We set $\delta_i$ to the standard deviation of $z_i$ over the collected data in our experiments. Nonetheless, we evaluate the sensitivity of HERON to these parameters in Figure 6. We find values in $[0, 2 * \sigma_i]$ work well, where $\sigma_i$ is the standard deviation of $z_i$.

**Tuning Cost.** Finally, we compare the tuning cost of

*Table 5.* Win rate against the SFT model as calculated by various LLM judges.

|  | RLHF-Flow | PairRM | Eurus-7B | Claude-3 | Avg. |
|---|---|---|---|---|---|
| Reward Eng. (Equal Weight) | 54.67 | 58.49 | 53.68 | 57.46 | 56.08 |
| Reward Eng. (Decaying Weight) | 59.24 | 59.64 | 52.49 | 56.85 | 57.06 |
| HERON-DPO | **66.20** | **63.02** | **63.22** | **63.82** | **64.57** |

*Table 6.* Ground-truth reward obtained in robotics environments.

|  | Ant | Hopper | Cheetah | Pendulum |
|---|---|---|---|---|
| Ground-truth | 0.99(0.0) | 0.86(0.01) | 0.94(0.10) | 1.0(0.01) |
| Reward Eng. | 0.88(0.02) | 0.72(0.05) | 0.61(0.04) | 0.99(0.0) |
| HERON | **1.0(0.01)** | **0.78(0.04)** | **0.62(0.04)** | **1.0(0.0)** |

HERON with reward engineering in the traffic light control environment. For HERON we consider tuning with exact domain knowledge (the hierarchy is given) and with inexact domain knowledge (the top 3 elements are given but their order is not specified). For reward engineering we consider tuning with exact domain knowledge and with no domain knowledge (the reward weights do not have hierarchical structure). For the latter case we tune the weights with bayesian optimization. The results are shown in Figure 6c. We find that both versions of HERON significantly outperform reward engineering. Although bayesian optimization can train high performing policies, it requires 5 to 15 times the tuning iterations of HERON.

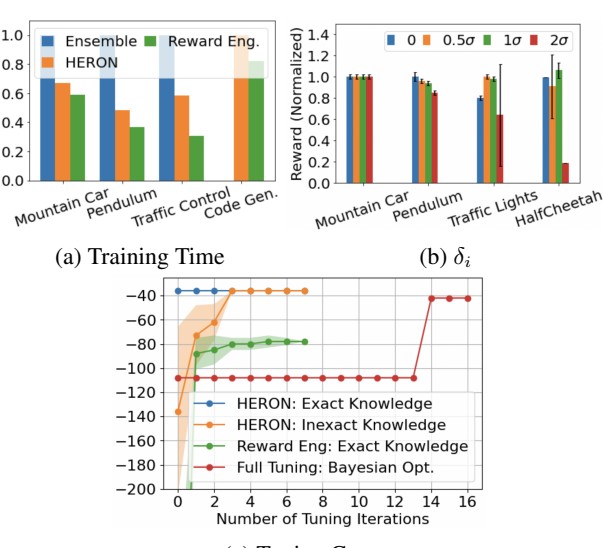

(a) Training Time          (b) $\delta_i$

(c) Tuning Cost

*Figure 6.* Training time and ablation study for HERON.

# 5. Discussion

**Suitable Scenarios.** HERON is not intended to serve as a solution for all RL problems; however, it can perform quite well in specific settings. In particular, HERON will be most useful in environments where there are several feedback signals available and a human overseer can rank these signals. Our experiments show that in such environments (code generation and traffic light control) HERON can outperform state-of-the-art baselines. Moreover, HERON shows great versatility as it can achieve decent performance in non-ideal environments (robotic control).

**Reward Flexibility.** HERON is also capable of dealing with the feedback signals that are of nearly equal importance. One feasible solution is to flip the importance ranking of the equally important feedback signals with certain probability during the preference elicitation step described in Section 3. One can even design more complex preference elicitation algorithms that do not require a strict hierarchy over feedback signals, which we leave for future work.

**Low Labeling Budget Setting.** RLHF is an effective method to obtain a powerful policy model. This is because humans can provide more informative insight for the reward model than feedback signals. However, involving humans to compare every pair of trajectories is often not affordable and it is difficult to create suitable instructions on how to compare trajectories. We consider a separate setting, where only some feedback signals and some domain knowledge about them are available. In this case, reward engineering becomes a reasonable and cheap choice, as does our method. Therefore, reward engineering is the most suitable baseline.

**Multi-objective RL.** Our method is not designed specifically for multi-objective RL problems. Generic multi-objective RL is complex because some objectives adversely affect other objectives during the optimization. In this case, researchers try to find the Pareto frontier and balance among the objectives in many different scenarios (Van Moffaert & Nowé, 2014; Mossalam et al., 2016). In contrast, our method can be applied if and only if the feedback signals are available and have hierarchal structure.

**Future Work.** Our proposed hierarchical comparison procedure enjoys flexibility and can be extended. For instance, we can consider the level of the feedback in the hierarchy as the preference strength. More specifically, preference outcomes drawn based on more important feedback make stronger preferences between two RL trajectories. To exploit such preference strength, we can add rescaling or margin hyperparameters to reward learning. We leave this for future investigation.

## Impact Statement

This paper presents work whose goal is to advance the field of Machine Learning. There are many potential societal consequences of our work, none which we feel must be specifically highlighted here.

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

## A. Appendix / supplemental material

## B. Classic Control Experiment Details

For the classic control experiments we use the OpenAI gym (Brockman et al., 2016). To train all policies we use the DDPG algorithm, where the policies are parameterized by three layer MLPs with 256 hidden units per layer. We use the Adam optimizer, and search for a learning rate in $[1 \times 10^{-5}, 1 \times 10^{-3}]$.

For mountain car we train for a total of 15000 timesteps and begin training after 5000 timesteps. For pendulum, we train for a total of 50000 timesteps and begin learning after 25000 timesteps.

## C. Baselines

### C.1. Ensemble Baseline

Beyond the ground-truth reward, we compare the HERON algorithm with two ensemble baselines inspired by Brys et al. (2017). These ensemble baselines train a separate policy on each feedback signal, and then combine the policies' outputs in a given state to select an action. In every environment we train each policy in the ensemble with the similar parameters as used for the reward engineering baseline and we again tune the learning rate in $[1 \times 10^{-5}, 1 \times 10^{-3}]$.

As described in the main text, we consider two variants of this ensemble based algorithm: one where the action is selected according to an average over each policy ($a \leftarrow \text{argmax}_{a \in \mathcal{A}} \sum_{k=1}^{n} \frac{1}{n} \pi_k(s, a)$) and one where the preference ranking used as input to HERON is used to combine the actions ($a \leftarrow \text{argmax}_{a \in \mathcal{A}} \sum_{k=1}^{n} \gamma^k \pi_k(s, a)$). With the second variant, $\gamma$ is selected from $\{0.25, 0.35, 0.45, \cdots, 0.95, 0.99, 1\}$.

### C.2. Reward Engineering Baseline

We also examine the performance of a reward engineering baseline where the reward is formulated as $\sum_{i=1}^{n} \beta^i z_i$, where $\beta$ is a hyperparameter selected from $\{0.3, 0.4, ..., 0.9, 1.0\}$ and $z_i$ are the normalized feedback signals. The feedback signals are ordered according to the HERON reward hierarchy, making this a very realistic and competitive reward engineering baseline. However, we came across a few challenges when trying to make this algorithm work. First, the feedback signals all need to be normalized, which either requires complex algorithms or multiple agent rollouts before training. In addition, we find that this baseline is very sensitive to $\beta$ and therefore has a higher tuning cost. In addition, it can often not beat the performance of HERON. We plot the performance of the reward engineering baseline in Figure 7. Note that this plot shows performance over all of training, and HERON typically displays larger reward (comparatively) in the last stages of training.

As we can see from Figure 7, the reward engineering baseline requires extensive tuning to achieve good performance. In addition, the choice of normalization strategy is very important (Figure 7f). These results further show the benefits of HERON.

## D. Robotics

All of our experiments are conducted with the PyBullet simulator (Coumans & Bai, 2016). The feedback signals in each environment are as follows: for Ant, it is whether the robot is alive, the progress towards the goal state, whether the joints are at their limits, and whether the feet are colliding. For HalfCheetah, the signals are the potential and the power cost. For Hopper, the signals are the potential, an alive bonus, and the power cost.

## E. Traffic Light Control

In our experiments we train four agents in a two by two grid. The length of each road segment is 400 meters and cars enter through each in-flowing lane at a rate of 700 car/hour. The traffic grid can be seen in Figure 8. The control frequency is 1 Hz, i.e. we need to input an action every second. The reward is based on the following attributes for each agent $n$:

- $q^n$: The sum of queue length in all incoming lanes.

- $wt^n$: Sum of vehicle waiting time in all incoming lanes.

- $dl^n$: The sum of the delay of all vehicles in the incoming lanes.

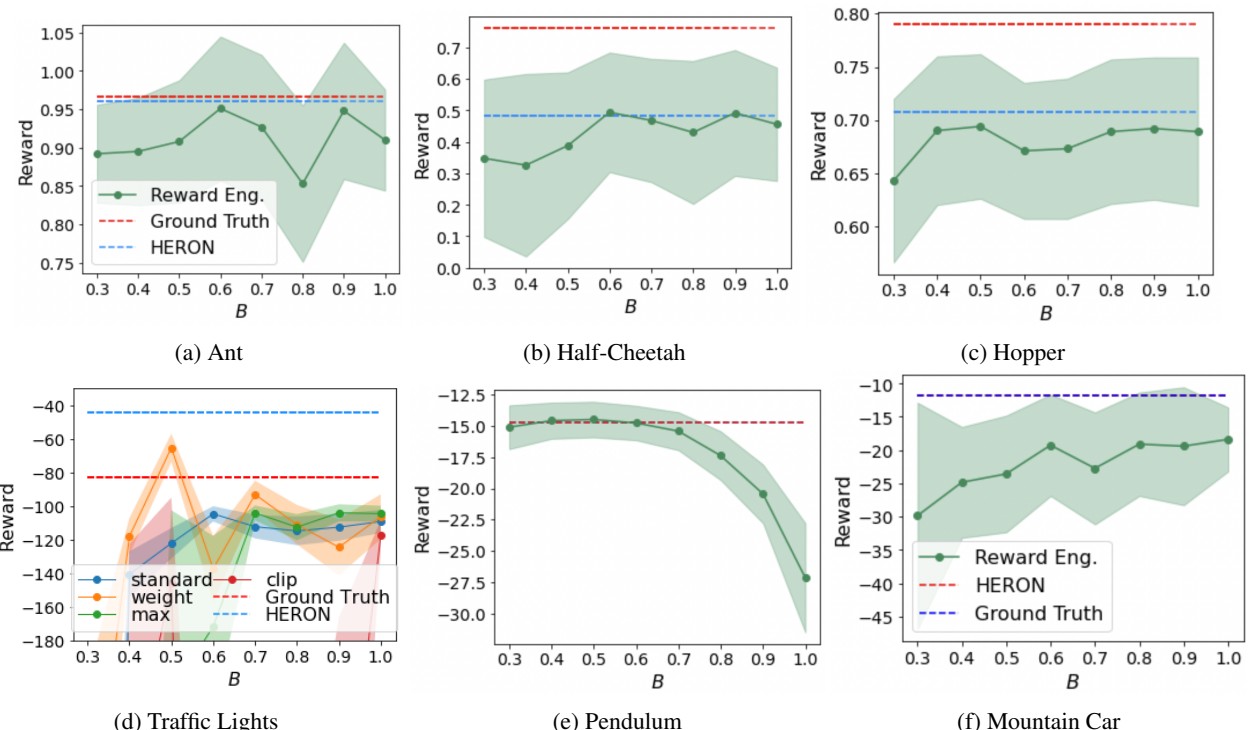

*Figure 7.* Ablation study of the reward engineering baseline.

- $em^n$: The number of emergency stops by vehicles in all incoming lanes.

- $fl^n$: A Boolean variable indicating whether or not the light phase changed.

- $vl^n$: The number of vehicles that passed through the intersection.

We can then define the reward-engineering reward as

$$R^n = -0.5q^n - 0.5wt^n - 0.5dl^n - 0.25em^n - fl^n + vl^n.$$

All algorithms have the same training strategy. Each agent is trained for three episodes with 3000 SUMO time steps each. At the beginning of training the agent makes random decisions to populate the road network before training begins. Each algorithm is evaluated for 5000 time steps, where the first 1000 seconds are used to randomly populate the road. For adversarial regularization, we use the $\ell_2$ norm to bound the attacks $\delta$.

## F. RLHF Comparison

To explicitly compare RLHF with HERON, we compare the algorithms in the pendulum environment. To simulate human feedback, we rank one trajectory over another if the ground truth reward achieved by that trajectory is higher than the ground truth reward achieved by the other trajectory. We then evaluate the performance of this simulated RLHF algorithm when varying amounts of feedback are given. The results can be seen in Figure 9. In this table we vary the number of feedbacks in RLHF, while keeping the number of feedbacks for HERON constant. In this setting HERON can perform as well as RLHF, but such good performance is not guaranteed in every environment.

## G. HERON Flexibility

In this section we evaluate how the behavior of the policies trained by HERON change when we change the reward hierarchy. We plot several hierarchies in Figure 10. The reward engineering is the thick black line. We try three signals as the most important signal (num_passed, wait time, and delay). We notice that all these observations can outperform the reward



*Figure 8.* Traffic light control environment.

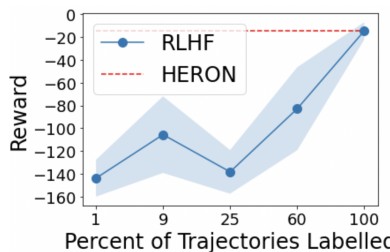

*Figure 9.* RLHF comparison in the Pendulum Environment.

engineering reward, even though we measure the return with the reward engineering reward. One important deviation from this good performance is when wait time is not ranked highly. The wait time is a very important signal, and when we do not put this variable high up in the hierarchy, the performance becomes unstable when measured according to the reward engineering reward. This is because if we ignore the wait time of cars, the policy may make some cars wait for a long time, which is not ideal. However, this can easily be accounted for in the reward design process.

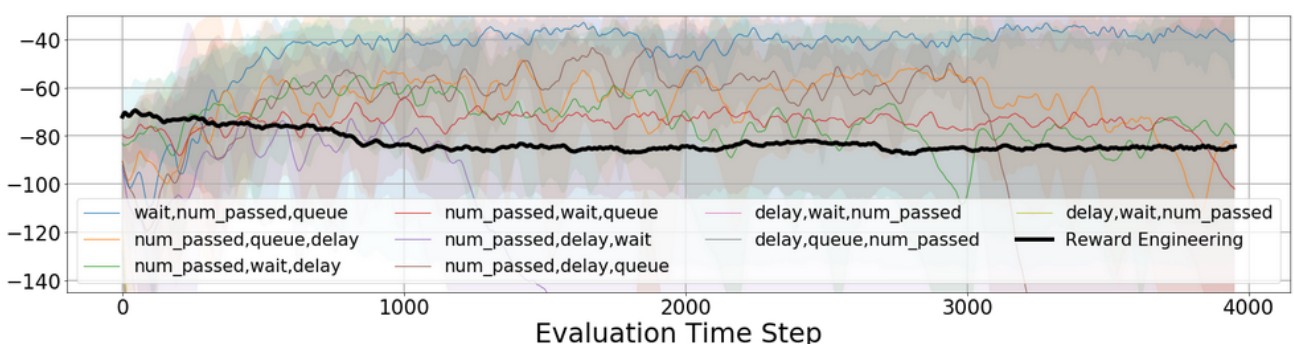

*Figure 10.* Different reward hierarchies in HERON.

## H. Code Generation

In this section we describe details for the code generation task.

### H.1. Behavior Cloning

To train the initial behavior model we use behavior cloning (supervised fine-tuning) to adapt the pre-trained CodeT5 to the APPS task. In particular, we use train with the cross-entropy loss for 12000 iterations, using a batch size of 64. We use the

Adam optimizer with a learning rate of $2 \times 10^{-5}$.

## H.2. Temperature Selection

A hyperparameter that can have a large impact on generation quality is the temperature parameter, which essentially alters how greedy we are in the next-token sampling step. In all settings we follow the implementation of Le et al. (2022), using a temperature of 0.6 for APPS and 1.2 for MBPP. In addition, we sample tokens greedily to construct a baseline sample for each problem.

## H.3. Reward Model

It has been noted that reward models often overfit to the dataset (Ouyang et al., 2022). Therefore we use a smaller version of CodeT5 for our reward model with only 220 million parameters. We train this model for around 40000 steps with a batch size of 64. This is roughly a single epoch on the preference dataset, which is comprised of 20 samples per problem sampled from the behavior model and some expert samples provided by the APPS dataset. We use the Adam optimizer with a learning rate of $2 \times 10^{-5}$.

## H.4. Reinforcement Learning

Once we have trained the reward model, we assign a reward to each program in our preference dataset and train using reinforcement learning on this dataset. Similar to Le et al. (2022), we train on the policy gradient loss and add the cross entropy loss as a regularization term. We compare our method to two reward engineering rewards:

**CodeRL reward.** The first reward we compare HERON to is from CodeRL, which defines the reward as

$$R_{\text{CodeRL}}(s) = \begin{cases} -1.0 & \text{if program } s \text{ fails to compile} \\ -0.6 & \text{if program } s \text{ has a runtime error} \\ -0.3 & \text{if program } s \text{ fails a unit test} \\ 1.0 & \text{if program } s \text{ passes all unit tests.} \end{cases}$$

**PPOCoder reward.** The second reward we compare HERON to is based on PPOCoder, which has the insight to include syntactic similarity to expert samples in the reward. This effectively smooths the reward, and can therefore make the reward more informative. In particular, they compare the abstract syntax trees of the generated programs with the expert example programs. This is computed as

$$R_{\text{ast}}(s, \widehat{s}) = \text{Count}(\text{AST}_s, \text{AST}_{\widehat{s}}) / \text{Count}(\text{AST}_s).$$

We then construct the final PPOCoder based reward as $R_{\text{PPOCoder}}(s) = R_{\text{CodeRL}}(s) + \lambda \text{MEAN}_{\widehat{s}}(R_{\text{ast}}(s, \widehat{s}))$, where $MEAN$ is the mean operator. We tune $\lambda \in \{0.001, 0.01, 0.1, 1\}$. We remark that the original PPOCoder reward contains more feedback signals, but we do not use all of them due to the large tuning cost required to tune the ourselves.

For both of these rewards and the HERON reward we tune the learning rate in $\{3 \times 10^{-6}, 5 \times 10^{-6}, 8 \times 10^{-6}\}$.

## H.5. Example Programs

To further analyze the performance of HERON, we examine some of the programs generated by HERON. These programs are randomly selected. We display concatenated prompts and completions in Figure 11.

# I. Reward Training

In this section we detail our reward model training. For the classic control tasks and the traffic light control task we do not have a good initial behavior policy, so we must train our reward model in an iterative manner. In these settings, we iteratively update the reward model using samples from the current version of the policy. In this way the reward model is trained on samples generated from progressively better policies.

As we mentioned in our discussion on the computational costs of HERON, the cost of reward model training depends on the frequency at which the reward model is trained. For the classic control environments we simply use a linear training

schedule, in which the reward model is updated every 400 steps. For traffic light control we train the reward model with an annealed frequency, where the reward model is trained every $100v^t$ steps, where $v$ is set 1.3 and $t$ is the current time step.

We demonstrate the multi-step reward model training in Figure 12. The sharp drop in accuracy occurs at time step 1000, where the behavior model changes from random to a trained policy. This large change in accuracy indicates that multi-step reward model training is needed, as reward models trained on random behavior do not perform as well when the behavior changes.

### I.1. The $\alpha$ Hyperparameter

**Formal description of shaping signal:** Given a trajectory $\tau$, let us compare it with $n$ other trajectories $\tau_1, \ldots, \tau_n$. Let $F(\tau)$ denote the average level of the decision tree $\tau$ wins at. To allow us to incorporate domain knowledge into HERON, we multiply the reward assigned to $\tau$ by a signal $\alpha^{F(\tau)}$, where $\alpha$ is a hyperparameter. When the feedback signals are categorical, $F(\tau)$ can capture which category $\tau$ lies in, and multiplying the reward by $\alpha^F(\tau)$ can control the reward separation between different categories.

**Visual description of shaping signal:** As mentioned in the main text, the $\alpha$ hyperparameter can be used to control the shape of the rewards. In Figure 13, we show how changing $\alpha$ changes the reward shape in the code generation task.

## J. Computational Setup

For the classic control tasks and traffic light control experiment we run experiments on Intel Xeon 6154 CPUs. For the code generation task, we train with Tesla V100 32GB GPUs.

## K. Robotics Learning Curves

In Figure 14 we display the learning curves in the robotics environments.

## L. Limitations

The main limitation of HERON is that not every problem will contain an obvious ranking over the feedback signals, as some signals may be equally important. We propose to mitigate this limitation in future works by allowing for ties or using a randomized decision tree in the preference elicitation procedure.

## M. Alignment Experiments

Here we present more details on our language model alignment experiments. We use LoRA (Hu et al., 2020) for all experiments. For the SFT base model, we train for two epochs with learning rate 5e-5. We use batch size 32 and train for 2 epochs. For Reinforce we also use learning rate 5e-5, batch size 32, and train for 2 epochs. For DPO, we use learning rate 5e-5, batch size 32, $\beta = 0.1$, and train for 2 epochs.

For evaluation, we use each reward model as specified in their respective release. For Claude 3 based evaluation, we prompt it to select the most correct, helpful, and harmless response.

```
You are an expert Python programmer, and here is your task: Write a function to find the list with minimum length us
ing lambda function. Your code should pass these tests:

assert min_length_list([[0], [1, 3], [5, 7], [9, 11], [13, 15, 17]])==(1, [0])
assert min_length_list([[1,2,3,4,5],[1,2,3,4],[1,2,3],[1,2],[1]])==(1,[1])
assert min_length_list([[3,4,5],[6,7,8,9],[10,11,12],[1,2]])==(2,[1,2])
```

```python
class Solution:
    def min_length_list(self, *invars):
        """
        :type invars: List[List[int]]
        :rtype: (int, List[int])
        """
        res, index = float('inf'), []
        for item in invars:
            if item == float('inf'):
                return 1, index
            elif not res:
                index = index + [len(item)]
            else:
                if index[-1] < len(item):
                    res, index = len(item), index + [len(item)]
                else:
                    index = index + [len(item)]
        return res, index
```

```
You are an expert Python programmer, and here is your task: Write a function to find the nth tetrahedral number. You
r code should pass these tests:

assert tetrahedral_number(5) == 35.0
assert tetrahedral_number(6) == 56.0
assert tetrahedral_number(7) == 84.0
```

```python
class Solution:
    def tetrahedral_number(self, *n):
        print(n)

        def f(x, x1, x2, y):
            if y < 0: return -1.0
            x, y = x + x1, y + y1
            if ((x > 5e-6 or y > 5e-6) and (x - x1 <= y - x2)): return -1.0
            if (y == 5e-6 and x1 < y - x2) or y == 5e-7 and x1 == x2: return -1.0
            z = 0
            while (x > x1 <= x2):
                if (x % 2 == 0): y -= 1
                x = x // 2
            while (y > y1 <= y2):
                if (y % 2 == 0): x -= 1
                y = y // 2
            while (z > abs(y1 - x1) > abs(y2 - x2)):
                z = abs(y1 - x1)
                x1, x2, y1, y2 = x2, x1, y2, z * (-1)
            return z * y

        a, b, w = *n
        p, q = 1, 3
        while True:
            x, y = a * q + x, b * w + y
            z = f(x, y, a * q, b * w + (a + 1) * q) if n else -1.0
            if z < 0.0:
                return y
            if (a % 2 == 0): y = y - 1
            a = a // 2
            if (q < 2): a = p * q + q
            if (p > 1.1e-6 and q > 5.6e-7): p = 1.1e-5
            if (w > 1.2e-6): w = 1.2e-6
            q = p * q + p
```

```
You are an expert Python programmer, and here is your task: Write a function to convert the given decimal number to
its binary equivalent. Your code should pass these tests:

assert decimal_to_binary(8) == '1000'
assert decimal_to_binary(18) == '10010'
assert decimal_to_binary(7) == '111'
```

```python
class Solution:
    def decimal_to_binary(self, x):
        """
        :type x: int
        :rtype: str
        """
        a = [0]*8;
        for i in range(8):
            a[i]+=1<<i;
        return ''.join(map(str, a));
```

*Figure 11.* Example programs generate by LLMs trained with HERON.

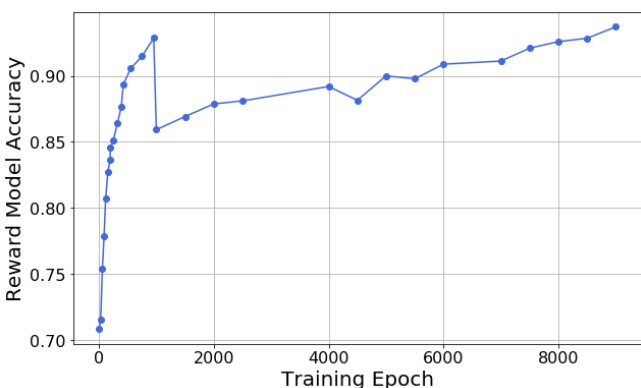

*Figure 12.* Reward model accuracy throughout training.

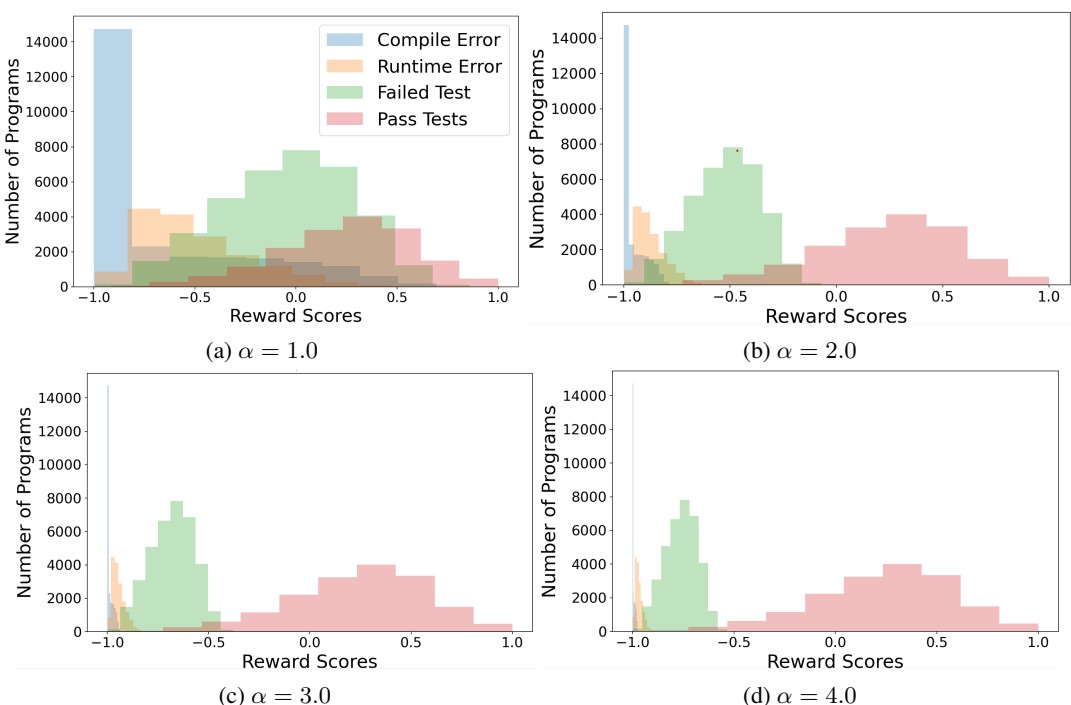

*Figure 13.* Reward shape with different values of $\alpha$.

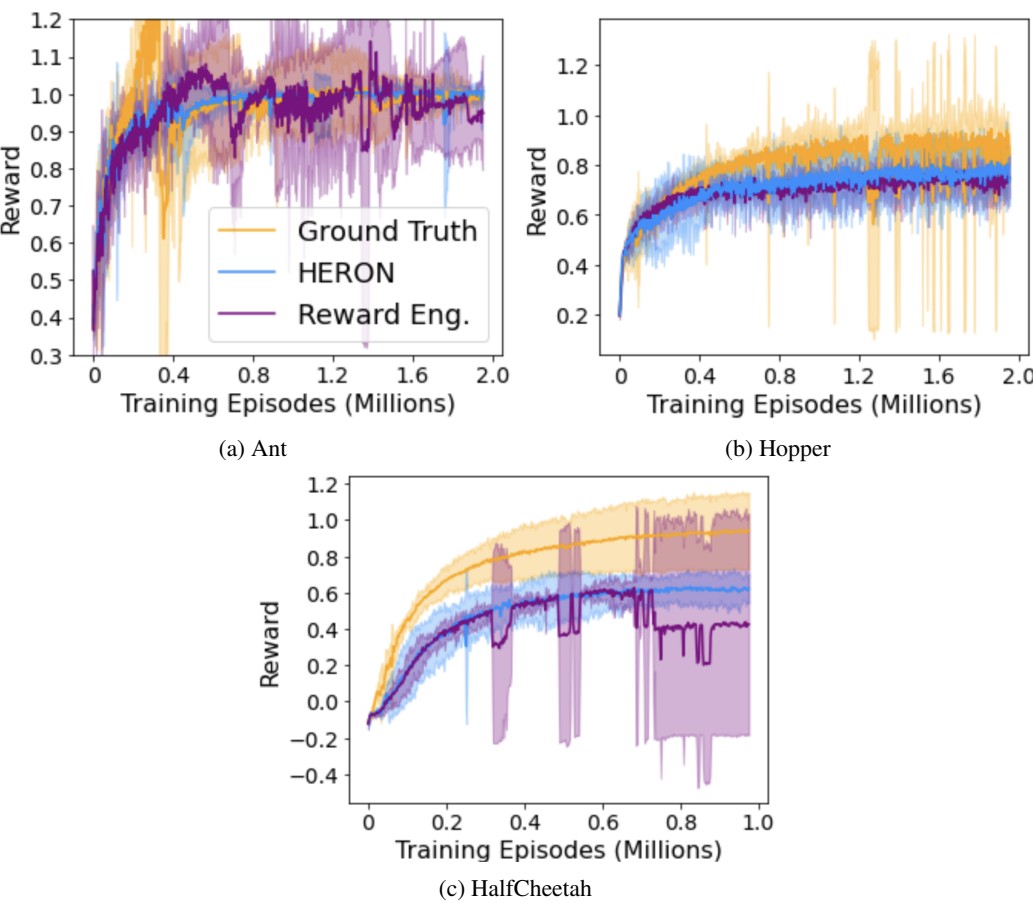

(a) Ant

(b) Hopper

(c) HalfCheetah

*Figure 14.* Training curves in different robotics tasks.

