# OpenReview forum: "Deep Reinforcement Learning from Hierarchical Preference Design"
_ICML.cc/2025/Conference — ICML 2025 poster_

### Official Review · Reviewer_Q7xT · 2025-02-24

**Overall Recommendation:** 3

**Summary:**

This paper proposes HERON, a novel hierarchical reward design framework for reinforcement learning (RL) that leverages the hierarchical structures of feedback signals to ease the reward design process. HERON constructs a decision tree based on the importance ranking of feedback signals to compare RL trajectories and trains a reward model using these comparisons. The authors demonstrate HERON's effectiveness across various RL applications, including traffic light control, code generation, language model alignment, and robotic control. In traffic light control, HERON outperforms reward engineering techniques and achieves higher performance than the ground-truth reward. In code generation, HERON surpasses state-of-the-art methods using handcrafted piece-wise reward functions, showing improved sample efficiency and robustness.

**Claims And Evidence:**

The claims made in the submission are generally supported by evidence. The authors provide extensive experimental results across multiple domains (traffic light control, code generation, language model alignment, and robotic control) to demonstrate the effectiveness of HERON compared to existing methods. However, the claim that HERON can achieve decent performance even in environments with unclear hierarchy (robotic control) is less convincing, as the results show mixed performance and the environments tested may not fully represent the complexity of real-world scenarios where hierarchy is unclear.

**Essential References Not Discussed:**

No

**Experimental Designs Or Analyses:**

The experimental designs and analyses presented in the paper are generally sound and well-structured. The authors conducted extensive experiments across multiple domains, including traffic light control, code generation, language model alignment, and robotic control, to validate the effectiveness of the HERON framework. However, the experiments could benefit from additional ablation studies to further isolate the impact of specific components of the HERON framework, such as the hierarchical decision tree and the preference-based reward model, on the overall performance.

**Methods And Evaluation Criteria:**

The proposed methods in the paper are well-suited for the problem of reward design in reinforcement learning, especially in scenarios where feedback signals have a natural hierarchy or where rewards are sparse. The evaluation criteria and benchmark datasets used, such as traffic light control, code generation, and robotic control tasks, are relevant and effectively demonstrate the versatility and effectiveness of the HERON framework. However, the evaluation could be further strengthened by including additional real-world applications with more complex hierarchical structures to better validate the robustness of HERON in diverse environments.

**Other Comments Or Suggestions:**

No.

**Other Strengths And Weaknesses:**

The paper's strengths include its original approach to reward design through hierarchical structures, which is a creative combination of existing ideas in preference learning and RL. The application of HERON to real-world use cases like traffic light control and code generation demonstrates its practical significance and potential impact. The clarity of the paper is also commendable, with well-organized sections and clear explanations of the methodology and results. However, a potential weakness is the limited exploration of scenarios with unclear hierarchical structures, which could restrict the framework's applicability in more complex real-world environments.

**Questions For Authors:**

1. Could the authors provide more details on how HERON handles scenarios where the hierarchy of feedback signals is not clear or where signals have equal importance? This would help in understanding the robustness of the framework in more complex real-world applications.

2. How does the computational cost of HERON compare to other state-of-the-art methods, especially in terms of training time and resource requirements? This information is crucial for assessing the practical feasibility of deploying HERON in resource-constrained environments.

3. Can the authors discuss the potential impact of HERON on multi-objective RL problems, where objectives may conflict? Understanding this could highlight the broader applicability of HERON beyond the current scope of the paper.

**Relation To Broader Scientific Literature:**

The key contributions of the paper are closely related to the broader scientific literature on reinforcement learning (RL) and reward design. Specifically, the proposed HERON framework builds upon prior work in hierarchical reward modeling and preference-based learning, offering a novel approach to leverage hierarchical structures in feedback signals to simplify reward design. This work also connects to recent advancements in deep RL, particularly in applications like traffic light control and code generation, where it demonstrates significant improvements over existing methods, highlighting its relevance and potential impact in the field.

**Theoretical Claims:**

The paper does not present any formal proofs for theoretical claims. Instead, it focuses on empirical validation through extensive experiments across various applications. Therefore, there are no proofs to verify for correctness in this submission.

---

> ### Author Rebuttal · Authors · 2025-04-01
>
> Dear Reviewer Q7xT,
>
> Thank you for the insightful review! We are glad you appreciate our work, and present our rebuttal to your review below.
>
> ## **Experimental Designs Or Analyses**
>
> **The experiments could benefit from additional ablation studies to further isolate the impact of specific components of the HERON framework, such as the hierarchical decision tree and the preference-based reward model, on the overall performance.**
> - Note that we conduct an extensive ablation study in section 4.5. We find that the margin parameter is quite important.
> - Furthermore, we also investigate the flexibility and robustness of HERON, finding HERON is both flexible and provides increased robustness.
>
> ## **Other Strengths And Weaknesses**
>
> **A potential weakness is the limited exploration of scenarios with unclear hierarchical structures, which could restrict the framework's applicability in more complex real-world environments**
> - First, we remark that HERON is designed for settings where there is a clear hierarchy (see our discussion on suitable scenarios) and performance in other settings is a “bonus.”  In settings with unclear hierarchies like robotic control (see section 4.4), we find HERON can still outperform reward engineering. Our intuition is that in these settings a roughly correct hierarchy provides sufficient information for good performance. If the signals have equal performance, we propose in line 402 to randomly flip the ranking of equal feedback signals. However, we leave this study for future work.
>
> ## **Questions**
>
> **Could the authors provide more details on how HERON handles scenarios where the hierarchy of feedback signals is not clear or where signals have equal importance?**
> - See our above response to **Other Strengths And Weaknesses**.
>
> **How does the computational cost of HERON compare to other state-of-the-art methods, especially in terms of training time and resource requirements?**
> - We show the training time of HERON versus baselines in Figure 6a. HERON is around 25% slower than reward engineering, but greatly reduces tuning cost (Figure 6c). HERON is faster than other baselines such as the ensemble baselines as well.
>
> **Can the authors discuss the potential impact of HERON on multi-objective RL problems, where objectives may conflict?**
> - Thank you for the suggestion. As we mention in Line 420, HERON is not designed for multi-objective RL (MORL). In fact, HERON and MORL are solving different problems. MORL tries to train a policy on the pareto frontier among several reward factors. In contrast, HERON tries to find a way to combine feedback signals into a reward in a user-friendly way, such that the user can easily guide the agent’s behavior. This is useful in many real-world tasks—like code generation, aligning language models, or traffic light control—where some goals are naturally more important than others. To make this clearer, we’ll move the MORL discussion to the related work section and explain how it differs from our approach.
>
>
> Thank you again for the insightful review, and please let us know if you have any other questions. We look forward to further discussion!

---

### Official Review · Reviewer_zX5S · 2025-02-25

**Overall Recommendation:** 4

**Summary:**

This paper introduces HERON, a decision-tree-based approach for reward design in reinforcement learning. In particular, the authors leverage human expertise to define a hierarchy of feedback signals. The authors employ that hierarchy to compare trajectories, collecting a dataset (with a policy model) and learn a reward model (similar to standard procedure in preference-based RL methods). The reward model is then used to train the policy, and the whole procedure can be repeated to improve the reward model. The authors extensively evaluate their approach in diverse scenarios (multi-agent traffic light control, code generation, language model alignment and standard control environments), highlighting its versatility and performance. Finally, the authors present an ablation study, focused on the training time, hyperparameter sensitivity and tuning cost.

**Claims And Evidence:**

The authors mostly position their claims of HERON over: (i) the overall performance of the method in comparison with baselines, (ii) the robustness of the method to changes in magnitude of the underlying reward signals, and (iii)  the adaptability of the method across multiple domains.

The empirical evidence presented in Section 4 supports many of these claims: the authors show that HERON outperforms relevant baselines across multiple settings. In Section 4.1, the authors also show how the performance of HERON is less sensitive to changes in the dynamics of the environment. The authors also present an extensive ablation study in Section 4.5, which provides additional insights on the sensitivity of the model to hyper parameters, as well as the training and computational cost of training HERON.

I would like to point out that in Section 4.2 (Results), the authors hypothesize that HERON's reward function "may be more conducive to learning". However no empirical support to this claim is provided, such as learning curves showing faster convergence.

**Essential References Not Discussed:**

A major missing reference and baseline is PEBBLE [1], a literature-standard method in preference-based RL. A direct comparison, in particular for the control experiments of Section 4.4., would clarify whether HERON’s benefits stem from its hierarchical structure or simply from being a preference-based reward modeling approach.

[1] Lee, Kimin, Laura Smith, and Pieter Abbeel. "Pebble: Feedback-efficient interactive reinforcement learning via relabeling experience and unsupervised pre-training." arXiv preprint arXiv:2106.05091 (2021).

**Experimental Designs Or Analyses:**

The experimental design and analysis appears to be sound. In particular, I would like to highlight the impressive range of evaluation scenarios employed in this work: from multi-agent systems, to code generation, LLM alignment and control tasks. Across all of them, HERON either outperforms or performs on par with the baseline methods.

However, I would like to single out HERON’s post-training reward scaling procedure in Section 4.2. This method appears to be an additional form of reward shaping, but it is not really motivated in the paper. The authors should experimentally evaluate if this extra step actually contributes to the performance of HERON, with an ablated version of the model.

**Methods And Evaluation Criteria:**

The proposed method is sound and addresses the real-world RL challenge of designing effective reward functions. The use of hierarchical preferences over reward features, instead of linear combinations is an interesting contribution and is well motivated and explained in the paper. The evaluation criteria are also appropriate for each domain: reward of the agents in the evaluation environment (given by a ground-truth reward function), win rates for language model alignment and pass@K metrics for code generation.

**Other Comments Or Suggestions:**

There is a typo in Page 2: "humans typically start with the the most important factor"

**Other Strengths And Weaknesses:**

One of the main strengths of this paper is that it presents an easily interpretable approach to reward design: the authors already explore some interpretability in Section 4.1 (Signal Utilization). The extensive evaluation suite is also to be commended.

However, a significant limitation of HERON is its assumption of an underlying ranking of feedback signals. While the authors evaluate HERON on control tasks where it is not clear what in the ranking, it still remains unclear if the level of performance observed would generalize to other tasks.

**Questions For Authors:**

1 - Can the authors justify better the need for the post-training reward scaling (presented in Section 4.2)? Shouldn't the original decision tree already learn to differentiate between these different feedback signals? Does HERON still outperform baselines without this heuristic adjustment?

2- How does HERON handle cases where the importance ranking of feedback signals is uncertain (for example in the experiments of Section 4.4)? Would its performance degrade significantly if the ranking were shuffled?

3 - Can you provide some empirical support to the claim that HERON's reward function "may be more conducive to learning" in Section 4.2? For example, showing the learning curves of HERON against the baselines.

**Relation To Broader Scientific Literature:**

The paper positions HERON within the area of preference-based RL. The authors discuss in Section 2 the connections and differences between the proposed method and others in RLHF, reward shaping, and inverse RL. The authors propose a decision-tree-based approach to compare agent trajectories to build a reward function. Tree-based structures for the reward function have been explored previously explored [1], but the use of a hierarchy of feedback signals appears to be novel. There is also a significant connection to multi-objective RL (MORL) literature, yet the authors only mention this at the end of the paper. I believe that it should deserve a section in related work, especially as the feedback signals employed by the authors can be considered different objectives in MORL.


[1] Bewley, Tom, and Freddy Lecue. "Interpretable Preference-based Reinforcement Learning with Tree-Structured Reward Functions." Proceedings of the 21st International Conference on Autonomous Agents and Multiagent Systems. 2022.

**Theoretical Claims:**

There are very little theoretical claims in this paper. The loss function of HERON (Equation 2) follows standard preference-based RL formulations and appears correct. There are no proofs in the paper.

---

> ### Author Rebuttal · Authors · 2025-04-01
>
> Dear Reviewer zX5S,
>
> Thank you for your detailed review and insightful suggestions! We are glad you think our work is novel. We provide our response to your review below. We shorten some questions to save space.
>
> ## **Claims and Evidence**
>
> **The authors hypothesize that HERON's reward function "may be more conducive to learning". However no empirical support to this claim is provided.**
> - We would like to clarify that this is an intuition rather than a definitive claim. Our hypothesis—that HERON's reward function may be more conducive to learning—was meant as a possible explanation for HERON's empirical performance advantage, not a conclusive statement. This hypothesis stems from the fact that many raw feedback signals are discrete or sparse, and reward models can serve as a way to smooth out this sparsity.
> - We do find some supporting evidence in Figure 14, where HERON exhibits significantly smoother and more stable learning curves compared to reward engineering. This suggests that HERON's reward function facilitates more stable training dynamics, which aligns with our intuition.
>
> ## **Experimental Designs Or Analyses**
> **However, I would like to single out HERON’s post-training reward scaling procedure in Section 4.2. This method appears to be an additional form of reward shaping, but it is not really motivated in the paper.**
> - Our post-training scaling is motivated by Figure 13, in which we found that reward modeling alone does not perfectly separate correct code from incorrect code at a global level. In order to further separate the two cases, we introduce a scaling parameter. Note that we only have a single parameter to tune, compared to the 4 used in CodeRL.
> - The pass@50 results without reward scaling can be found below. HERON still performs decently (outperforming all baselines, which are carefully tuned in prior works), but we find that post-scaling can further improve performance. We will include the results in the appendix.
> ### Table: Pass@50 Scores
> | Model       | Pass@50 Score |
> |-------------|---------------|
> | HERON       | 9.85          |
> | CodeRL      | 9.81          |
> | HERON+PTS   | 10.19         |
> | PPOCoder    | 7.62          |
> | BC          | 6.74          |
>
>
> ## **Relation To Broader Scientific Literature**
>
> **There is also a significant connection to multi-objective RL (MORL) literature...**
> - Thank you for pointing out reference [1], we will include it in our related work. As for MORL,we remark that MORL tries to find the pareto frontier among several reward factors. In contrast, HERON tries to find a way to combine feedback signals into a reward in a user-friendly way, such that the user can easily guide the agent’s behavior. We believe these are two orthogonal directions. We will move discussion of MORL to the related work section.
>
> ## **Essential References Not Discussed**
>
> **A major missing reference and baseline is PEBBLE**.
> - PEBBLE is not directly comparable to HERON, as PEBBLE relies on human preference data for learning. HERON on the other hand is meant for settings where we try to design a reward function from some freely available feedback signals. Therefore PEBBLE cannot be used as a baseline in our experiments.
> - Moreover, PEBBLE’s contributions (unsupervised pre-training and off-policy learning) are orthogonal to HERON’s contributions. Therefore we do not include PEBBLE as a reference.
>
> ## **Weaknesses**
>
> **However, a significant limitation of HERON is its assumption of an underlying ranking of feedback signals.**
> - While HERON assumes an underlying ranking of feedback signals, this assumption is well-motivated by many real-world scenarios where feedback signals naturally have a hierarchy—for example, in traffic light control, code generation, and LLM alignments. In such contexts, ranking the relative importance of feedback signals is valid. We believe good results in these environments already represent a significant contribution.
> - Moreover, as demonstrated in Section 4.4, HERON performs well even in the absence of a strict feedback hierarchy. This empirical resilience highlights its practical utility and broad applicability, even when the ranking structure is noisy or only partially specified.
>
> ## **Questions**
> **Q1:** See above response to **Experimental Designs Or Analyses**.
>
> **Q2:** We find HERON can still perform well in these settings, outperforming reward engineering baselines (Table 6). HERON is relatively robust to shuffled rankings. In Figure 6, we show how HERON performs with inexact domain knowledge, i.e. only knowing which factors fall in the top 3 and which ones fall in the bottom 3. By one tuning iteration (i.e. the best of two shuffled rankings out of a possible 36), HERON can already outperform the best-case performance of reward engineering.
>
> **Q3**: See above response to **Claims and Evidence**.
>
> Thank you again for your thoughtful review, and please let us know if you have any further questions.

---

> > ### Comment · Reviewer_zX5S · 2025-04-02
> >
> > I would like to thank the authors for answering my questions. For this reason, I increase my score.

---

> > > ### Author Response · Authors · 2025-04-02
> > >
> > > Dear Reviewer zX5S,
> > >
> > > Thank you for the quick response and willingness to increase your score. We will be sure to include the discussed modifications into the next version of our paper.

---

### Official Review · Reviewer_7gbW · 2025-03-15

**Overall Recommendation:** 3

**Summary:**

The paper proposes HERON, a hierarchical preference-based RL framework that leverages the hierarchical importance of feedback signals to design reward functions. HERON constructs a decision tree based on human-provided importance rankings of feedback signals to compare trajectories and train a preference-based reward model. The authors claim HERON improves sample efficiency, robustness, and applicability to sparse reward settings. Experiments are conducted on traffic control, code generation, and other tasks to validate these claims.

**Claims And Evidence:**

The claim that HERON universally eases reward design is problematic since many real-world tasks do not naturally offer clearly prioritized feedback signals. More comprehensive experimental validation and discussion are needed to convincingly support the broad claims.

**Essential References Not Discussed:**

Lack of disscussions of recent studies, such as [1], [2].

[1] Hejna, J., & Sadigh, D. (2023). Inverse preference learning: Preference-based rl without a reward function. Advances in Neural Information Processing Systems, 36, 18806-18827.

[2] Verma, M., & Metcalf, K. (2024) Hindsight PRIORs for Reward Learning from Human Preferences. In The Twelfth International Conference on Learning Representations.

**Experimental Designs Or Analyses:**

The choice of baselines is relatively outdated, lacking comparisons with recent advancements such as inverse preference learning approaches (e.g., Hejna & Sadigh, 2023) and Hindsight PRIORs (e.g., Verma & Metcalf, 2023).It remains unclear if the experiments cover a sufficiently diverse range of scenarios, especially those where feedback signals do not exhibit a clear hierarchical structure.

**Methods And Evaluation Criteria:**

**Methods**:

The use of pairwise comparisons to train a reward model is innovative, yet its dependency on accurate, human-specified importance rankings may limit its scalability and applicability in domains lacking clear hierarchy.

**Evaluation Criteria**:

The benchmarks and tasks selected for evaluation appear to showcase improved sample efficiency and robustness. However, the evaluation does not sufficiently address scenarios where the hierarchical structure is ambiguous or entirely absent, raising concerns about the method’s generalizability.

**Other Comments Or Suggestions:**

No

**Other Strengths And Weaknesses:**

**Strengths**:

Innovative hierarchical framework that intuitively aligns with human decision processes.

Empirical results indicating improvements in sample efficiency and robustness on the tested tasks.

**Weaknesses**:

Reliance on a clear hierarchy in feedback signals, which may not be present in many practical applications.

Lack of theoretical analysis to ensure that the learned reward function leads to optimal or near-optimal policies.

Experimental comparisons are limited to older baselines, missing insights from more recent literature.

**Questions For Authors:**

1.	How does HERON perform in environments where feedback signals do not have an inherent hierarchical structure?

2.	Have you considered leveraging neural network architectures to automatically learn or approximate the hierarchical priorities when explicit human rankings are unavailable?

3.	Can you provide any theoretical analysis or formal guarantees regarding the convergence and optimality of the policies derived from the HERON framework, especially in sparse reward settings?

4.	What is the rationale behind the selection of baseline methods, and how do you expect HERON to compare against more recent state-of-the-art methods like those proposed by Hejna & Sadigh (2023) and Verma & Metcalf (2024)?

5. How sensitive is HERON to noisy or suboptimal human rankings of feedback signals?

[1] Hejna, J., & Sadigh, D. (2023). Inverse preference learning: Preference-based rl without a reward function. Advances in Neural Information Processing Systems, 36, 18806-18827.

[2] Verma, M., & Metcalf, K. (2024) Hindsight PRIORs for Reward Learning from Human Preferences. In The Twelfth International Conference on Learning Representations.

**Relation To Broader Scientific Literature:**

The work builds upon prior research in preference-based reinforcement learning and reward shaping by introducing a hierarchical structure that mimics human decision-making. A more comprehensive discussion of how HERON fits into and advances the current state-of-the-art would be beneficial.

**Theoretical Claims:**

**Theoretical Analysis**:

The paper does not provide a rigorous theoretical analysis or proofs ensuring that the learned reward function preserves the optimal policy.

**Issues**:

Without theoretical guarantees, there is a risk that the policy may converge to a suboptimal local optimum, particularly in sparse reward environments where reward signals are less distinct.

---

> ### Author Rebuttal · Authors · 2025-04-01
>
> Dear Reviewer 7gbW,
>
> Thank you for your detailed review and suggestions to improve our work. First and foremost we would like to address your review of our claims, in which you say we claim “HERON universally eases reward design”. Nowhere in the paper do we make this claim, and in fact we consistently state that our scope is limited to problems with hierarchical structure (see line 51 right, line 62 left, and line 436 left).
>
> We present our rebuttal of the remainder of your review below. We shorten some questions to save space.
>
> ## **Experimental Design or Analyses**
>
> **The choice of baselines is relatively outdated, lacking comparisons with [1] and [2]**
> - We believe there is a misunderstanding here. HERON is not comparable to preference learning methods. HERON assumes online access to an environment which gives the agent several feedback signals. We seek to design a reward function from these feedback signals, which is a classical setting in RL. On the other hand, [1] requires access to an offline preference dataset, while [2] requires access to online preference annotation. In contrast, HERON asks a human annotator to rank the importance of feedback signals (usually there are 3-7 such signals) one time. This is completely separate from the recommended papers. We include a detailed discussion on suitable scenarios in line 436.
>
> **It remains unclear if the experiments cover a sufficiently diverse range of scenarios...**
> - We conduct extensive experiments in 7 diverse environments, which we believe is sufficient to show the benefit of HERON (multi-agent traffic light control, code generation, LLM alignment, and four robotic control experiments). In robotic control experiments the feedback signals do not all have a clear hierarchy, yet we are able to beat or match the baselines in all environments. These results demonstrate that our method performs well in a diverse set of hierarchical or roughly hierarchical environments.
>
> ## **Relation To Broader Scientific Literature**
> **A more comprehensive discussion of how HERON fits into and advances the current state-of-the-art would be beneficial.**
> - Thank you for the suggestion. We will add more works on SOTA approaches in our related works section, including [1] and [2]. Please note we already have a discussion on suitable settings for HERON (line 436).
>
> ## **Weaknesses**
> **Weakness 1**
> - First, we want to point out that HERON is designed for settings where there is a clear hierarchy (see our discussion on suitable scenarios) and performance in other settings is a “bonus.” However, in environments without clear hierarchy, we find HERON can still perform decently, outperforming reward engineering baselines. See section 4.4 of our paper for more details.
>
> **Weakness 2**
> - Theoretical analysis is challenging, as it is difficult to precisely characterize the relationship between the feedback signals and the ground truth reward. For example, in traffic light control, defining an optimal combination of all relevant factors is inherently ambiguous.
> - That said, we hypothesize that if theoretical guarantees were to be established, the convergence of the policy would likely depend on two key factors: (1) the statistical error in the learned reward, and (2) the distribution shift between the state visitation distributions of the sampling policy and the optimal policy.
>
> ## **Questions**
> **Q1:** See our response to weakness 1.
>
> **Q2:** One approach is to learn a decision tree based on human preference data. This may work when the feedback signals have hierarchical structure and we only have a limited amount of human preference data. However, this would be a separate setting and we leave it for future work.
>
> **Q3:** See response to weakness 2.
>
> **Q4:** The main baseline we compare against is reward engineering, which has been the most popular approach for reward design over the past 20 years. We also compare against two baselines that have been used in MORL literature, the ensemble baselines. In relevant settings such as code generation, we compare to carefully designed and publicized reward functions like that of CodeRL. Finally, when available, we also compare training directly on the ground-truth reward. Again we note that those SOTA methods of [1] and [2] are not relevant in our setting, as they assume access to a large set of human preference data (either online or offline) which HERON is not designed to use.
>
> **Q5:** HERON is relatively robust to suboptimal rankings. In Figure 6, we show how HERON performs with inexact domain knowledge, i.e. only knowing which factors fall in the top 3 and which ones fall in the bottom 3. By one tuning iteration, HERON can already outperform the best-case performance of reward engineering. This indicates HERON can perform well even with slightly noisy rankings.
>
> Thank you for the detailed review, and please let us know if you have any further questions or need any clarification.

---

> > ### Comment · Reviewer_7gbW · 2025-04-02
> >
> > Thanks for the response that addresses my concerns.

---

> > > ### Author Response · Authors · 2025-04-02
> > >
> > > Dear Reviewer 7gbW,
> > >
> > > Thank you for the rebuttal acknowledgment and the willingness to raise your score. We will include the modifications we discussed in the next version of our paper.

---

### Official Review · Reviewer_uDMx · 2025-03-20

**Overall Recommendation:** 4

**Summary:**

In this work, the authors propose a novel hierarchical reward design framework tailored for environments that require the integration of multiple feedback signals. The framework is motivated by the observation that these signals often contribute unequally to the overall reward and that there is a hierarchy in how much each feedback contributes. The paper includes extensive empirical evaluations across four different applications and compares the proposed method with alternative reward design approaches.

---

## update after rebuttal

Thank you to the authors for their detailed response. After reading the rebuttal and considering the other reviews, my main concerns were addressed, and I changed my original score.

---

**Claims And Evidence:**

The paper’s claims are primarily empirical—for instance, the authors claim improvements in robustness and overall performance compared to existing techniques in specific scenarios. Although the results appear to support these claims, crucial details are missing from the empirical evaluation section, making it difficult to fully confirm that the evidence supports the claims. Please refer to the detailed comments and questions in the Empirical Designs or Analyses section of the review.

**Essential References Not Discussed:**

The reference list appears comprehensive, and I did not identify any essential references that were missing.

**Experimental Designs Or Analyses:**

The experimental design is thorough, but several details require further clarification:
- For the traffic light control experiment:
    - How many samples were used? The plots show considerable variance, and the sample size is important for drawing reliable conclusions.
    - What is the definition of the ground truth reward, and why does it show more variance compared to the other methods? Additionally, why do the other reward design methods appear to perform better than it?
    - Could the authors clarify why it is important that a relatively small proportion of decisions are made at each level (as suggested by the results in Figure 3), and how these proportion values were computed?
     - In the experiment presented in Figure 4, why was only the second feedback signal varied? What difference would it make if the number of cars passed (the first feedback) was not kept constant?
     - Could the authors elaborate on the setup for the robustness experiment detailed in Figure 5? For instance, what was the original training speed, to which speed was it changed afterward, and how many trials were used?
- For the code generation experiments, what is the number of trials used, and what is the variance in the results? While the paper claims that the proposed method “significantly outperforms the baselines,” some results (e.g., those in Tables 1-4 for HERON and CodeRL) are fairly close, which might challenge this claim if the variance is high.
- In the LLM alignment experiments, why was HERON-DPO compared with REINFORCE? Could the reward engineering baseline potentially perform better if coupled with a different algorithm (e.g., PPO or TRPO)?
- Regarding the robotics experiment, could the authors clarify the details of Table 6? For example, what do the numbers represent, how were they generated (e.g., the number of trials), what is the final hierarchy used for the feedback signals, and how was this hierarchy determined?

**Methods And Evaluation Criteria:**

The method is generally well-described and motivated. The evaluation is also thoughtfully executed: the authors selected a variety of environments with distinct characteristics to showcase the different properties of the proposed method. For example, they demonstrate how the method performs when there is a clear hierarchy in the feedback signals versus when such a hierarchy is absent.

**Other Comments Or Suggestions:**

Figure 6 is missing axis information.

**Other Strengths And Weaknesses:**

The method is clearly described and straightforward, making it easy to follow. The motivation is also well-articulated and intuitive. However, as noted in earlier sections of the review, there are important aspects of the empirical evaluation that should be further clarified in the paper.

**Questions For Authors:**

1. On line 176, the authors mention that "it is possible to introduce pre-trained knowledge into the reward model." Could the authors provide concrete examples of this?
2. On line 198, the authors state that "in appropriate settings, we can use DPO …". Could the authors clarify what constitutes an appropriate setting in this case? For instance, are there particular types of tasks or domain characteristics that make DPO especially suitable?

**Relation To Broader Scientific Literature:**

This paper introduces a novel method for combining multiple feedback signals into a single reward measure. The approach offers advantages over traditional techniques, such as linear combinations with engineered weights, particularly in scenarios where the feedback signals have a clear hierarchy. Although the method addresses a somewhat specific problem, it appears to perform well in that context. I believe that the community could benefit from this approach.

**Theoretical Claims:**

The paper does not present any theoretical claims or proofs.

---

> ### Author Rebuttal · Authors · 2025-04-01
>
> Dear Reviewer uDMx,
>
> Thank you for your thoughtful and detailed review of our paper. We are glad you appreciate the novelty of our approach as well as our experimental analysis. We provide our rebuttal to your critiques below. We enumerate your questions and comments to save characters.
>
> ## **Experimental Design and Analysis:**
>
> ### **1. Traffic Light Control:**
>
> **1.1: How many samples were used?**
> - We present results over five random seeds. Indeed we find that the baselines have high variance, but HERON exhibits very low variance. To further validate the efficacy of our method, we have run a t-test for the last 1000 evaluation time steps in the traffic light control environment. Our method is significantly better than the reward engineering baseline, with p=0.003. This gain justifies the use of our algorithm.
>
> **1.2: Ground truth reward.**
> - The ground truth reward can be found on line 693, and it has been developed over several papers. It was finalized in [Zhang 2019].  Other baselines sometimes outperform it, as we extensively tune all baselines. This tuning can be seen as a type of reward shaping, which can improve performance.
>
> **1.3: Proportion of decisions made at different levels.**
> - If too many decisions are made at a single level, that effectively means that information from other levels of the decision tree are being completely ignored, which is not ideal. Decisions being made at all levels indicate information from all feedback signals is being incorporated into the reward, which is desirable and indicates the efficacy of our reward design. We compute the proportions in traffic light control by recording which level of the tree each decision was made at throughout training.
>
> **1.4: Figure 4.**
> - We only vary the second feedback signal to keep the reward realistic, as the first feedback (traffic throughput) is almost always viewed as most important in traffic light control. We show the policy can still achieve good performance with other hierarchies in Figure 10.
>
> **1.5: Figure 5.**
> - The original speed is 35, then we change it to 25, 30, 40, 45 (See Figure 5). We conduct 5 independent trials for each experiment and report mean and standard deviation over the runs.
>
> ### **2. For the code generation experiments, what is the number of trials used, and what is the variance in the results?**
> - We conduct training over 1 seed due to the large expense of these training runs, which is in line with prior works.  However, when evaluating a policy on APPS, we evaluate each policy with 1 million generated programs (there are 5000 test problems, we generate 200 for each problem) and 100000 on MBPP. Treating each set of programs as an independent Bernoulli variable, we can conduct a t-test. When considering the largest value of K in pass@k in tables 1, 2, 3, 4 we find that the a t-test comparing HERON and the best baseline has p values < 0.05, indicating that HERON indeed outperforms the baselines in a statistically significant manner.
>
>
> ### **3. In the LLM alignment experiments, why was HERON-DPO compared with REINFORCE?**
> - We compare with REINFORCE as it is one of the most popular and high performing approaches for LLM alignment these days [1]. It is possible the baseline could do better with PPO, but this would require extensive tuning and PPO is shown to underperform REINFORCE for LLM alignment [1].
>
> [1] Ahmadian, Arash, et al. "Back to basics: Revisiting reinforce style optimization for learning from human feedback in llms."
>
>
> ### **4. Regarding the robotics experiment, could the authors clarify the details of Table 6?**
> - In table 6 we show the average ground truth reward obtained by each algorithm over the final 1000 iterations of training (training is 2 million steps total). We conduct experiments over 5 random seeds, and use the PPO algorithm to optimize the policies. The reward hierarchy is generally is-alive > moving forward > control cost > contact cost. We will make this more clear in the paper.
>
> ## **Questions**
>
> **Q1**: In code generation, we use a pre-trained language model as the initialization for the reward model. This is advantageous as it means reward training is faster and more accurate. Similarly for LLM alignment we could use a pre-trained LM as a reward, but opt for DPO in our experiments due to its simplicity.
>
> **Q2**: DPO is most useful when the horizon length of the task is one, as in that case we can directly train the policy on preference comparisons, without having to train a reward model. This is the case in LLM alignment, which is where DPO is mainly used.
>
> Thank you again for the detailed review. We are eager to know if your questions about our empirical evaluation have been satisfied, or if there are any more details we can give you.

---

### Decision · Program_Chairs · 2025-05-01

**Decision:**

Accept (poster)

**Comment:**

The paper introduces HERON, which leverages the rankings of multiple reward factors to derive reward functions using a hierarchical decision tree.

Strengths

* Interpretability of proposed framework in reward modeling

* Extensive evaluation across various domains (multi-agent traffic light control, code generation, language model alignment and standard control environments)

Weaknesses

* Assumption of an underlying ranking of feedback signals

Overall, all reviewers agreed that this is a very solid submission and the authors also handled concerns from reviewers during the discussion period. I think the paper makes a nice contribution that the community will find valuable.